# Effect of Alpha-S1-Casein Tryptic Hydrolysate and L-Theanine on Poor Sleep Quality: A Double Blind, Randomized Placebo-Controlled Crossover Trial

**DOI:** 10.3390/nu14030652

**Published:** 2022-02-03

**Authors:** Kokila Thiagarajah, Huei Phing Chee, Nam Weng Sit

**Affiliations:** Department of Allied Health Sciences, Faculty of Science, University Tunku Abdul Rahman, Bandar Barat, Kampar 31900, Malaysia; cheehp@utar.edu.my (H.P.C.); sitnw@utar.edu.my (N.W.S.)

**Keywords:** alpha-s1-casein tryptic hydrolysate, L-theanine, sleep, cortisol, alpha power

## Abstract

This randomized, placebo-controlled, crossover and double-blind study investigates the effects of RLX2™ containing alpha-s1-casein tryptic hydrolysate and L-theanine in working adults affected by poor sleep quality. The supplement or placebo was randomly and blindly assigned to 39 subjects for four weeks and the changes in the subjective sleep assessment via the Pittsburgh Sleep Quality Index (PSQI), heart rate, blood pressure, salivary cortisol by high-performance liquid chromatography method and alpha power of awake electroencephalogram (EEG) were studied. The data were analyzed in two ways, by crossover and crossover summed up. The latter depicted that RLX2™ improved PSQI total score, sleep latency, sleep duration, sleep habitual efficiency, daytime dysfunction, and increased total and frontal alpha power significantly (*p* < 0.05). The supplement prolonged the total sleeping time by 45 min in the supplement receiving group compared to the placebo group (*p* < 0.001). However, only sleep duration and sleep habitual efficiency showed a profound effect in both analyses (*p* < 0.05). In conclusion, being given its beneficial effects without notable adverse events, it would be advantageous to use these nutraceutical ingredients to promote better sleep quality. Further studies with a larger number participants are warranted to support these findings.

## 1. Introduction

Healthcare professionals have stressed the importance of assessing sleep quality in addition to sleep quantity. Sleep quality is better related to sleepiness, health, and wellbeing than sleep quantity in a non-clinical population [1]. Meanwhile, poor sleep quality is a defining criterion of insomnia [2,3], and this sleep disorder may interfere with normal physical, mental and emotional functioning, yet affected people are often not aware of their problem and may not be getting appropriate treatment.

Few studies have been carried out to explore the epidemiological nature of sleep problems, particularly insomnia, among Malaysian populations. Nevertheless, a study conducted among 2049 adult primary care patients in Malaysia revealed that 60% had insomnia symptoms. It was stated that for those 50 years old above, anxiety and depression were risk factors for chronic insomnia with daytime dysfunction [4]. Insomnia may lead to anxiety and stress and vice versa which can be a risk factor for psychotic disorder. People with persistent sleep disturbances are more prone to accidents, have higher rates of work absenteeism, diminished job performance, decreased quality of life, and increased health care utilization [5].

A list of pharmacological treatment options such as benzodiazepines (e.g., estazolam, temazepam, and triazolam), nonbenzodiazepine hypnotic agents (e.g., eszopiclone, zaleplon, and zolpidem) and orexin receptor antagonists (e.g., suvorexant) are readily available for the treatment of insomnia. Unfortunately, they do have numerous side effects, such as dizziness, drowsiness, memory damage, cognitive impairment, dependence, tolerance, and withdrawal symptoms such as anxiety. Besides, patients might abuse the drugs, which could lead to injuries and accidents due to the sedation effect of the prescribed sleep medication [6,7,8]. These medications may also be inappropriate for those suffering from mild to moderate sleep disorders or acute insomnia. In addition, most of the affected people do not prefer a pharmacological approach for insomnia treatment [9] and often switch to alternative treatments such as nutritional supplements.

Alpha-s1-casein tryptic hydrolysate (CTH) is a unique bioactive decapeptide within a milk protein hydrolysate that has been claimed to have calming properties. Messaoudi et al., [10] showed that CTH can improve mental conflict and physical stress situations in study subjects. In the experimental stress test, consuming CTH could significantly reduce the systolic blood pressure, diastolic blood pressure and cortisol level. The anxiolytic-like effect of CTH could be due to the presence of α-casozepine, specifically the αs1-CN(f91–100) in it. This decapeptide serves as a positive modulator for the gamma-aminobutyric acid type A (GABA_A_) receptor. Similar to benzodiazepines, it increases the inhibition responses of GABA which can produce sedative, anxiolytic, and muscle relaxant effects. GABAergic transmission is being targeted for induction and maintaining sleep as a treatment for insomnia [11,12]. In a preclinical study, rats treated with 300 mg/kg/day of CTH had an increase in the GABA_A_ receptor activation and at the same time an increase in total sleep and a decrease in time awake were detected [13]. Meanwhile, a dosage of 300 mg/day of CTH for four weeks resulted in no adverse events in humans [14,15].

On the other hand, L-theanine is a type of amino acid found mainly in green tea. It is claimed to have anti-stress or relaxing properties, helps maintain normal sleep, improves memory function, neutralizes the effects of neurotoxin, ameliorates cardiovascular disease, and has other health benefits [16,17,18,19]. The amino acid able to initiate an alpha wave pattern in the brain implies a relaxed physical and mental state without drowsiness or impaired motor skills. Besides, it is involved in the formation of the inhibitory neurotransmitter GABA, which is similar to CTH [20]. In healthy subjects, the intake of L-theanine has been shown to reduce the heart rate and serum Immunoglobulin A which attenuate sympathetic nervous activation by blocking the binding of glutamic acid to the corresponding receptors in the brain [21]. In pre-clinical studies, L-theanine is neither mutagenic nor carcinogenic, and has low toxicity, as its median lethal dose (LD_50_) value is 5000 mg/kg while the no-observed-adverse-effect-level (NOAEL) for L-theanine is 4000 mg/kg/day [19,22].

Academic staff from a Malaysian university were randomly chosen in this study and screened for sleep quality by using the Pittsburgh Sleep Quality Index (PSQI), as it is a reliable screening tool with high specificity and sensitivity in differentiating good and poor sleep quality. In addition, all the components of PSQI were able to distinguish primary insomnia from a healthy population [2,23]. Knowing that sleeping problems can lead to stress or anxiety and vice versa, it can be postulated that the combination of CTH and L-theanine can be apt to relieve the symptoms. The two active ingredients were formulated into a single capsule, known as RLX2™ and produced by LiveLife Bioscience AG, Zurich, Switzerland. Those with poor sleep quality were randomly assigned in this double-blind, placebo-controlled, and crossover clinical trial to explore the effect of the supplement on sleep quality by assessing PSQI, blood pressure, heart rate, salivary cortisol, and alpha power. The sleep trial duration of four weeks proved to be more beneficial than two weeks [15]. Thus, four weeks of treatment with either the supplement or placebo were used in this study.

## 2. Materials and Methods

### 2.1. Investigational Product and Reagents

RLX2™ and the placebo were provided by LiveLife Biosciences AG, Zurich, Switzerland. The supplement contained 150 mg CTH and 50 mg L-theanine in the same capsule while the placebo was a low-fat milk powder, encapsulated similar to the supplement. The capsules were labelled as VC65003 and VC6500A and the content information of the capsules was revealed by a manager from LiveLife Sdn. Bhd., Malaysia after completion of the study period.

The following chemicals were obtained from the respective manufacturers: cortisol (11-α, 17-α, 21-trihydroxy, α-4 pregnene 3,20-dione, hydrocortisone; mw 362.5; purity 98%) from Nacalai Tesque, Kyoto, Japan; 6-α methylprednisolone (mw 374.5; purity 98%), which was used as an internal standard (IS), from Tokyo Chemical Industry, Tokyo, Japan; methanol HPLC grade and n-hexane HPLC grade (98%) from Rank Synergy (Synerlab), Kuala Lumpur, Malaysia; diethyl ether analytical grade (99.8%) from Ansan-si Gyeonggi-do, South Korea; acetonitrile HPLC grade (99.9%) and acetone HPLC grade (99.8%) from QRec (Asia), Rawang, Malaysia.

### 2.2. Study Design

Screening for sleep disorder/insomnia was conducted among randomly selected academic staff in Universiti Tunku Abdul Rahman (UTAR), Kampar, Malaysia. Subjects who scored more than 5 in PSQI, which indicates poor sleep quality [23], and could adhere to the appointment schedules were included in the study. Those who were pregnant, taking sleeping pills or any supplements to improve their sleep quality within the past month, consuming more than 10 cups of green tea, coffee, or tea in a day, are lactose intolerant, have a myocardial infarction, or liver and kidney disorders were excluded from the study. Written informed consent was obtained from all the selected subjects in accordance with the Declaration of Helsinki and Malaysian Good Clinical Practice Guidelines. All the subjects completed a questionnaire about their medical history.

By using G*Power software (3.1.9.7 version, Franz Faul, University of Kiel, Kiel, Germany) [24], it was estimated that a total of 30 participants would be required with an anticipated non-compliance or drop-out rate of 33%, yielding 80% power to detect an effect (Cohen’s d = 1.3099) based on previously published data [25] with a two-tailed *t*-test at a 0.05 significance level. In total, 42 subjects out of 143 affected respondents with poor sleep quality (PSQI > 5) were chosen. This clinical trial was a randomized double-blind, placebo-controlled and cross-over study of nine weeks in total, including four weeks of treatment (RLX2™ or placebo) and a week of washout period in between the treatments, as shown in Figure 1. The study was approved by the UTAR Scientific and Ethical Review Committee (U/SERCl66l20l7).

One participant was withdrawn as she became pregnant and another participant in Group A completed stage 1 but withdrew his consent during the washout period. During stage 2 of the same group, one participant was suspended as she travelled overseas during the period. The participants were instructed to orally take the capsules daily an hour before sleep and the compliance for study was checked verbally. PSQI score, blood pressure, heart rate, and awake electroencephalography (EEG) were recorded and saliva samples for cortisol analysis were collected before and after the treatment (baseline, week four, week six and week nine).

### 2.3. Psychological Measurement using PSQI

The participants were required to fill out the PSQI [26], before and after the treatment of each stage. The PSQI is a self-rated questionnaire that assesses sleep quality and disturbances over a month-long time interval. Nineteen discrete items of the questionnaire produce seven individual component scores: sleep quality, sleep latency, sleep duration, habitual sleep efficiency, sleep disturbances, use of sleeping medication, and daytime dysfunction. The sum of scores for these seven components (range of score 0–3) would generate a total score of subjective sleep quality.

### 2.4. Blood Pressure and Heart Rate Measurements

Blood pressure and heart rate of the participants were measured by using the blood pressure monitor Microlife BP3AG1 (Microlife AG, Widnau, Switzerland) and an Omron Heart Scan HCG-801 (Omron Healthcare, Kyoto, Japan), respectively, before and after each treatment.

### 2.5. Salivary Cortisol Analysis

#### 2.5.1. Sample Collection

The Salivette^®^ Cortisol (Sarstedt AG & Co., Nümbrecht, Germany) sampling device was used to collect saliva samples, as it is designed to achieve precise analytical values from small volumes and samples with a very low level of salivary cortisol, an indicator of chronic insomnia [27]. Participants were required to collect saliva within 30 to 45 min after awakening. A simple mouth rinse with water for at least 10 min was suggested before collecting the sample. The participants were not allowed to eat, drink, brush teeth, smoke or use tobacco products, use any mouthwash or engage in exercise or similar physical activity before collecting the sample. The saliva–cotton sample (about 2 mL obtained by chewing the cotton swab for 2 min) was directly spat without force or inducement into the internal vial of the double-chamber tube, capped and stored at +4 °C until transported to the laboratory. The samples were centrifuged at 2000× *g* for 10 min at room temperature to remove any particulates and the supernatant was aliquoted and stored at −80 °C for later analysis.

#### 2.5.2. Solid Phase Extraction

When ready to analyze, the aliquots were thawed and centrifuged again at 2000× *g* for 10 min at room temperature to break down mucins and to obtain a clear fluid. The supernatants were collected and poured into freshly labeled tubes. The extraction of saliva samples was carried out using the solid phase extraction (SPE) technique modified from Elio et al. [27]. SPE Discovery DSC-18 columns (1 mL, Supelco, Bellefonte, PA, USA) were used for saliva sample clean-up and enrichment. Prior to extraction, 500 µL of centrifuged saliva sample or cortisol standard (10, 20, 40, 60, 80, and 100 nmol/L) were spiked with 5 µL of IS (final concentration: 53 nmol/L). Before adding the sample, the SPE columns were equilibrated with 3 mL of methanol followed by 1.5 mL of deionized water. After sample introduction, the columns were washed sequentially with 0.25 mL of deionized water, 0.5 mL of acetone:water mixture (1:4, *v*/*v*), and 1 mL of hexane. Finally, 1.5 mL of diethyl ether was used to elute the sample. Each eluted sample was dried in a vacuum chamber and consecutively resuspended with 100 µL of HPLC mobile phase prior to injection.

#### 2.5.3. High-Performance Liquid Chromatography (HPLC) Analysis

The HPLC equipment (Shimadzu, Kyoto, Japan) was comprised of the solvent delivery module CBM-20A, the photo-diode array detector SPD-M20A, and column oven CTO-10AS VP. The analytical column used was a stainless-steel pentafluoro-phenyl-propyl silica column (15 cm × 2.1 mm, 5 µm) (Discovery, HS-F5, Supelco, Bellefonte, PA, USA). An isocratic elution using acetonitrile and water (27% and 73%) as the mobile phase was chosen to separate the analytes. The injection volume was 20 µL. The mobile phase flow rate was 0.25 mL/min and the column temperature was kept at 35 °C. The absorbance was measured at 244 nm. The chromatogram was acquired using the LCsolution software, Version 1.0.0.1 (Shimadzu, Kyoto, Japan). Each sample analysis was completed in 15 min. The standard calibration curve was plotted by using the peak height ratio of cortisol (standard) against IS and was used to calculate the amount of cortisol present in saliva samples [27].

### 2.6. Electroencephalography (EEG) Analysis

An Emotiv Epoc EEG wireless headset (Emotiv Systems Inc., San Francisco, CA, USA) was used to record EEG signals of participants at the same time that other data were collected in a quiet setting. The signals (with eyes open) were obtained over three min from 14 active (AF3, AF4, F3, F4, F7, F8, FC5, FC6, P7, P8, T7, T8, O1, and O2) and two reference (CMS and DRL) electrodes. The active electrodes were placed at different locations on the scalp (Figure 2) to detect the electrical activity of neurons. The signal from each electrode was sampled at a frequency of 128 Hz and converted to digital form with the aid of a built-in 16-bit analog to digital converter.

The recorded raw data were preprocessed with the help of EEGLAB software [29]. The data was loaded in the software and the linear Finite Impulse Response (FIR) filter was used to band pass the data between 1 and 60 Hz band decomposition. Next, the resulting data was plotted for visual inspection and the artefacted portion of the data was eliminated using the eegplot function. The clean EEG signal was decomposed into different frequency bands and power spectral analysis was carried out for the alpha band. The spectral analysis was performed using the spectopo function provided in the MATLAB software version r2020a signal processing toolbox associated with EEGLAB (with a resolution of 1 Hz) to measure absolute power density (*µ*V^2^/Hz). For the spectral analysis, the input data had to be stationary. Therefore, the EEG data was divided into brief epochs with 1000 ms each, thus in this small interval of time the data was considered stationary to perform the analysis. All the channels were used to calculate total alpha power while frontal alpha power was measured from the electrodes AF3, AF4, F3, F4, F7, F8, FC5, and FC6 and the average power was recorded. Due to artefacts and inconsistent EEG data, two participants were excluded from the study during the pre-processing.

### 2.7. Statistical Analysis

IBM SPSS Statistics for Windows, Version 26 (IBM Corp, Armonk, NY, USA) was used for data analysis. Comparisons between the parameters of the two groups were conducted using the paired *t*-test. All the tests were two-tailed and the level of statistical significance was set at *p* < 0.05.

## 3. Results

### Study Participants

The data collection was conducted from June 2017 to September 2018. A total of 20 male (51.3%) and 19 female (48.7%) participants were enrolled in the trial. Table 1 shows the age and baseline characteristics of the study participants. No adverse events were reported from any of the participants throughout the study period.

Table 2 shows the crossover variables after the supplement or placebo administration. Based on the PSQI total score, supplement receiving groups had significant improvements (*p* < 0.001) compared to placebo in both stages. Although the placebo receiving group in Stage 1 showed a significant decrease in the PSQI total score by 1.38 (*p* = 0.026), this is likely caused by a placebo effect. The number of hours of actual sleep was significantly improved only in the supplement receiving groups, where they were able to sleep 55 and 37 min longer (*p* = 0.003 and *p* = 0.002) than before the treatment.

In individual component analysis, the supplement effect over placebo was clearly seen in the components 3 and 4, sleep duration and sleep habitual efficiency whereby only supplement receiving groups showed significant improvements. Subjective sleep quality and sleep latency components showed significant improvements in both placebo and supplement receiving groups in Stage 1 whereas only the supplement receiving group was significantly improved in Stage 2. Although the placebo had a significant effect, the scores were not greater than the supplement receiving group. A significant reduction of daytime dysfunction (component 7) was only observed in the supplement receiving group in Stage 2 (*p* < 0.001). Component 6 was not included, as none of the participants took sleeping medications.

Neither blood pressure (systolic and diastolic) nor heart rate showed any significant changes between the supplement and placebo receiving groups. As for salivary cortisol levels, neither the supplement nor the placebo receiving group marked any significant differences in Stage 1. Meanwhile, more favorable results were obtained within group B in Stage 2, whereby the placebo receiving group showed no significant difference while the supplement receiving group recorded a significant decrease of cortisol levels of 34.6% (*p* = 0.007).

Based on EEG analysis, the total alpha power as well as the frontal alpha power were significantly (*p* = 0.031, 0.037) increased only in the supplement receiving group in Stage 1.

Table 3 shows the crossover summed up variables after the supplement or placebo administration. Overall, the supplement effects were much more convincing in this type of analysis. As for PSQI analysis, the total score and all the components except for sleep disturbances showed significant improvement (*p* ≤ 0.001) for the supplement receiving group. A significant placebo effect was noticeable only for the subjective sleep quality component (*p* = 0.037).

No significant changes were recorded for systolic and diastolic blood pressures, heart rates, and cortisol levels in neither the supplement nor the placebo receiving groups except that systolic blood pressure was slightly decreased in the placebo receiving group (*p* = 0.021). However, for the EEG analysis, both total alpha power and frontal alpha power were significantly improved in the supplement receiving group (*p* = 0.022 and 0.008) compared to the placebo group.

It is evident from both analyses (Table 2 and Table 3) that the sleep duration and habitual sleep efficiency were significantly improved in the supplement receiving group. Thus, this supplement clearly increased the total sleeping time by a mean of 45 min and the time spent in bed asleep without any adverse event being reported from the participants. Meanwhile, Component 6 (use of sleep medications) was not reported, as none of the participants were taking sleep medications.

## 4. Discussion

Given that hyperarousal may be exacerbated by many factors such as work or financial stress, the aim of this study was to investigate whether four weeks of oral daily intake of the supplement would improve various sleep factors in Malaysians suffering from poor sleep quality. Numerous studies were conducted either on CTH [15,30] or L-theanine [16,21,31,32], or with other supplement combinations [25], and they showed mixed effects. To our knowledge, this is the first clinical trial that studied the possible effect of the combination of CTH and L-theanine on alpha power and salivary cortisol concentration besides sleep quality. In addition, most of the intervention was conducted on clinical patients [14] or healthy participants [10,31,32]. Thus, this study may resemble a general population with a degree of variability in sleep patterns.

Our results were comparable to a trial using CTH alone (211.21 mg/day) for the Japanese insomniac participants whereby the most profound improvement was observed in subjective sleep quality after two weeks of treatment. The sleep latency and daytime dysfunction were decreased after four weeks while the sleep disturbances were only improved after 35 days. There was also a significant difference in the placebo group although CTH-treated participants marked more significant changes (up to *p* < 0.005) [30]. In another study using a sleep diary as a sleep subjective measurement, the total sleeping time, sleep latency, and sleep efficiency were improved significantly in the CTH treatment group (300 mg/day), while the placebo group showed a worsening in sleep quality [15].

The PSQI findings from the present study were consistent with the study conducted among a healthy population in Japan using L-theanine at a dosage of 200 mg/day for four weeks. It improved PSQI, depression, and anxiety scores compared to the placebo group. Some of the components of the PSQI scores, i.e., sleep latency, sleep disturbances, and the use of sleep medication decreased significantly compared to placebo. Additionally, L-theanine administration did improve cognitive functions, particularly verbal fluency and the executive function of the subjects [31].

A recent clinical trial using the same supplement was evaluated on Malaysian subjects with sleep disorders. A profound effect was observed in the intervention group in terms of sleep latency, sleep disturbances, and daytime dysfunction; furthermore, stress, anxiety and depression scores decreased significantly relative to the placebo group [33]. Studies reported that there were placebo effects comparatively with treatments of CTH (PSQI total score) [15] and the combination of CTH and L-theanine (lower sleep latency and habitual sleep efficiency) [33]. The placebo effect can occur in any components of PSQI since there was no specific trend. Thus, the placebo effect was not critically affecting the results compared to the supplement receiving group, which was more consistent with those published studies.

The study conducted by Messaoudi et al. [10] showed that in the healthy subjects treated with 400 mg of CTH for three times 12 h apart, their systolic and diastolic blood pressures recorded significantly lower percentage changes in the Stroop test (cognitive function), while only systolic blood pressure showed a significantly lower percentage change in a cold pressure test (cardiovascular response due to stress) compared to the placebo group. The CTH did not affect the heart rate as it remained stable in both tests, which is in agreement with our study. Phing and Chee [33] reported that there was a placebo effect whereby the diastolic blood pressure was significantly lower in both placebo and the supplement- treated groups. In contrast, our study revealed a similar placebo effect only for systolic blood pressure in the crossover summed up analysis.

Salivary cortisol level was associated with long term sleep problems where chronic insomnia symptoms predicted an abrupt increase of cortisol in the morning [34]. CTH with a dosage of 150 mg/day for 14 days was able to reduce salivary cortisol levels as well as stressful symptoms in healthy subjects [35]. CTH was also able to reduce blood cortisol concentration in another study. However, this was a short-term effect with a considerably higher dosage of CTH (three doses of 400 mg) that was given to healthy males who were subjected to mental and physical stress and studied within 36 h [10]. In contrast, the administration of L-theanine did not decrease salivary cortisol levels in the healthy population even given at a higher dosage (200 mg/day for four weeks), but it did improve sleep quality, anxiety, depression, and cognitive functions. It has been postulated that changes of cortisol levels are transient [31], whereby a study with a single dose of 200 mg L-theanine showed a significant decrease of salivary cortisol post 1 h of treatment in a moderately stress-induced healthy population compared to the placebo group [32].

In our study, salivary cortisol concentration decreased significantly in line with the daytime dysfunction of the supplement receiving group (B) (Table 2). Daytime dysfunction can be associated with less enthusiasm to perform tasks [36]. The salivary concentration of cortisol in the morning may range from 10.2–27.3 nmol/L for normal adults [37]. Based on the cross over summed up analysis, cortisol levels decreased and fell into the normal range, although the change was not significant in the supplement receiving group. In contrast, cortisol levels remained higher than the normal range in the placebo group.

The EEG results suggested that the participants treated with the supplement had a better quality of sleep. Subjective sleepiness is negatively correlated with the total alpha power of the cortical region in awake EEG [38]. Increased alpha power indicates augmentation of arousal. This will contribute to better performance and increased alertness, which could have been affected if a person had a lack of sleep regardless of whether they were suffering from sleep-disordered breathing [39]. Frontal alpha power was associated with creative cognition, indicative of a heightened internal awareness [40]. A single dose of 200 mg of L-theanine has been shown to increase total and frontal alpha power after 3 h of intake in a stress-induced healthy population, and this reflects its calming effect [32].

CTH alone with a dosage of 300 mg/day showed better efficacy when 62.5% of the outpatients that came for psychiatric consultation affected with poor sleep quality were able to engender significant clinical responses compared to 45% of patients who take CTH with standard treatments such as antidepressants and mood stabilizers. CTH produced double the anxiolytic and pro-hypnotic effects amongst the outpatients. However, this study was neither randomized nor double-blinded. Nevertheless, the absence of adverse events or side effects of CTH can improve the compliance among participants compared to standard drugs in open-label studies like this [14]. In a recent study, CTH promoted sleep better than the synthetic decapeptide, α-casozepine, as CTH has slower degradation and might have other sleep-enhancing properties [41]. Meanwhile, studies on L-theanine revealed that the anxiolytic effect can be achieved with a single dose of up to 200–250 mg per day without any sedative effect [19].

Similar to many other studies, there were no reported adverse events, indicating the safety or tolerability of four weeks of this supplement intake. Although some parameters exhibited a placebo effect, the significance levels were much lower compared to the supplement receiving group. Four weeks of intervention with a minimal dosage might not be sufficient to produce a significant response. Furthermore, data from this study support the translational value of these nutraceutical ingredients considering their apparent benefits. Further studies with extended washout periods with various dosages in comparison with a standard treatment are warranted to make the supplementary treatment approach more valid considering there was a lack of adverse events.

This trial was conducted in a real-world setting and might be influenced by biases such as the consumption of beverages other than tea and coffee, diet, and stress. The study was also carried out on a small population of 39 people, which may make it subject to type II errors, although the sample size had been estimated according to the effect size from the previous study in patients with poor sleep quality [25]. Furthermore, the study might not represent the general population of Malaysia and its wide range of social backgrounds, as it was conducted among the academic staff of an institute.

## 5. Conclusions

This study was performed among Malaysian adults who were randomly screened from a working population and recruited in a randomized, placebo-controlled, double-blind crossover trial to explore the efficacy of a daily supplement for four weeks. The regimen significantly improved PSQI total score, sleep latency, sleep duration, sleep habitual efficiency, daytime dysfunction, and alpha power in tested groups. The results suggest that the supplement can mitigate poor sleep quality and may also have an anxiolytic effect. Nonetheless, the present report may pave the way for future investigation amongst specific populations like the elderly, adolescents, pregnant or lactating women or patients with comorbidities with different dosages, especially higher dosages for L-theanine and in comparison to the individual supplement (CTH alone and L-theanine alone).

## Figures and Tables

**Figure 1 nutrients-14-00652-f001:**
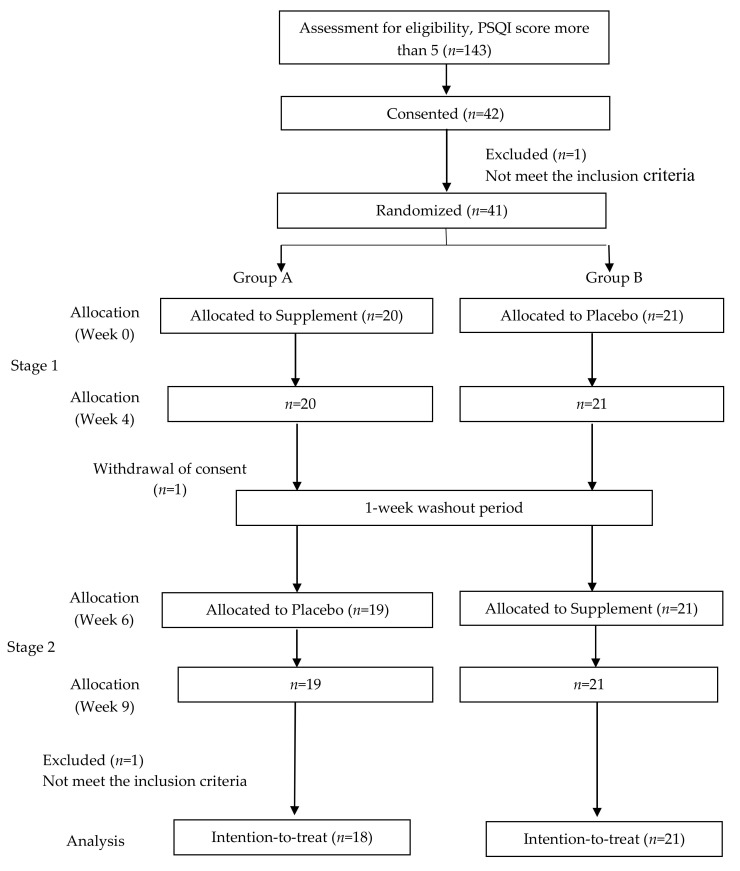
Study protocol.

**Figure 2 nutrients-14-00652-f002:**
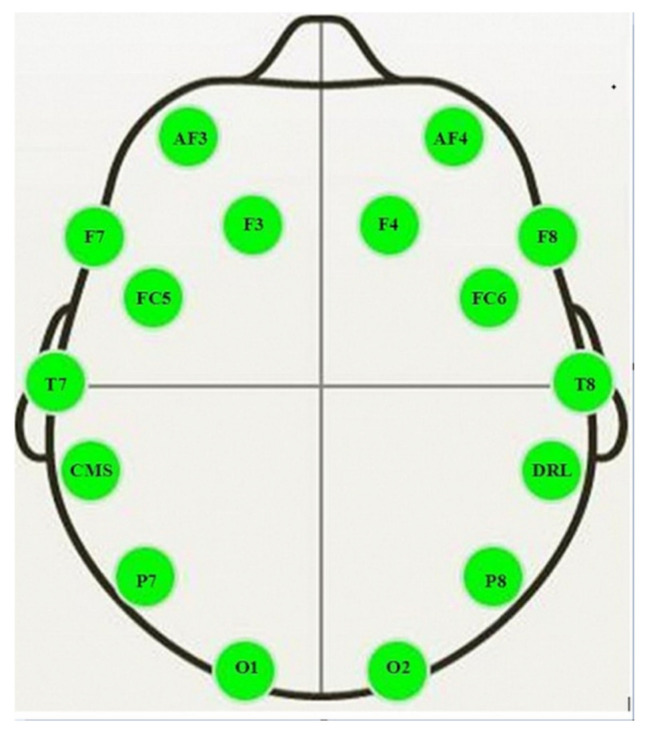
Locations of the active electrodes (AF3, AF4, F3, F4, F7, F8, FC5, FC6, P7, P8, T7, T8, O1 and O2) and reference electrodes (CMS and DRL) on the scalp for Emotiv EPOC headset [28].

**Table 1 nutrients-14-00652-t001:** The age and baseline characteristics of the study participants.

Variables	Mean ± SD (*n* = 39)	Range
Age (years)	37.3 ± 8.4	26–65
Systolic blood pressure (mmHg)	117.92 ± 13.33	91–144
Diastolic blood pressure (mmHg)	75.82 ± 11.86	51–102
Heart rate (beats per min)	77.54 ± 8.26	63–100

**Table 2 nutrients-14-00652-t002:** Crossover variables after the supplement RLX2™ or placebo administration.

Primary Outcomes	Group	Treatments	Pre Mean ± SD	Post Mean ± SD	*p*-Value
PSQI Total Score					
Stage 1	Group A	Supplement	9.00 ± 2.27	5.83 ± 1.54	<0.001 ***
	Group B	Placebo	7.90 ± 1.37	6.52 ± 2.40	0.026 *
Stage 2	Group A	Placebo	5.56 ± 2.33	5.67 ± 2.40	0.834
	Group B	Supplement	6.76 ± 2.39	4.95 ± 2.48	<0.001 ***
Total Sleeping Time (h)					
Stage 1	Group A	Supplement	5.28 ± 1.08	6.19 ± 0.69	0.003 **
	Group B	Placebo	5.71 ± 1.02	6.21 ± 1.27	0.061
Stage 2	Group A	Placebo	5.97 ± 0.95	6.11 ± 0.95	0.514
	Group B	Supplement	6.10 ± 1.28	6.71 ± 1.35	0.002 **
Component 1 Subjective Sleep Quality					
Stage 1	Group A	Supplement	1.72 ± 0.46	1.11 ± 0.58	0.004 **
	Group B	Placebo	1.57 ± 0.60	1.24 ± 0.70	0.049 *
Stage 2	Group A	Placebo	1.17 ± 0.62	1.06 ± 0.64	0.430
	Group B	Supplement	1.38 ± 0.50	1.05 ± 0.74	0.016 *
Component 2 Sleep Latency					
Stage 1	Group A	Supplement	1.67 ± 0.97	1.00 ± 0.77	0.004 **
	Group B	Placebo	1.48 ± 0.68	1.10 ± 0.83	0.029 *
Stage 2	Group A	Placebo	0.94 ± 1.06	0.94 ±1.00	1.000
	Group B	Supplement	1.24 ± 0.94	0.90 ± 0.94	0.005 **
Component 3 Sleep Duration					
Stage 1	Group A	Supplement	1.89 ± 0.68	1.22 ± 0.43	0.001 **
	Group B	Placebo	1.52 ± 0.97	1.24 ± 0.89	0.137
Stage 2	Group A	Placebo	1.28 ± 0.75	1.39 ± 0.70	0.495
	Group B	Supplement	1.33 ± 0.97	0.95 ± 0.80	0.008 **
Component 4 Sleep Habitual Efficiency					
Stage 1	Group A	Supplement	1.06 ± 1.00	0.17 ± 0.38	0.003 **
	Group B	Placebo	0.67 ± 0.73	0.52 ± 0.87	0.576
Stage 2	Group A	Placebo	0.33 ± 0.49	0.44 ± 0.62	0.542
	Group B	Supplement	0.62 ± 0.86	0.33 ± 0.66	0.030 *
Component 5 Sleep Disturbances					
Stage 1	Group A	Supplement	1.56 ± 0.51	1.39 ± 0.70	0.269
	Group B	Placebo	1.48 ± 0.51	1.33 ± 0.48	0.329
Stage 2	Group A	Placebo	1.17 ± 0.62	1.06 ± 0.54	0.495
	Group B	Supplement	1.10 ± 0.30	1.05 ± 0.50	0.666
Component 7 Daytime Dysfunction					
Stage 1	Group A	Supplement	1.11 ± 0.58	1.00 ± 0.49	0.331
	Group B	Placebo	1.14 ± 0.48	1.10 ± 0.77	0.789
Stage 2	Group A	Placebo	0.72 ± 0.57	0.78 ± 0.73	0.668
	Group B	Supplement	1.14 ± 0.73	0.67 ± 0.58	<0.001 ***
Systolic Blood Pressure (mmHg)					
Stage 1	Group A	Supplement	118.61 ± 12.50	121.56 ± 12.24	0.330
	Group B	Placebo	117.33 ± 14.30	112.52 ± 19.07	0.117
Stage 2	Group A	Placebo	123.11 ± 14.44	119.17 ± 11.95	0.077
	Group B	Supplement	116.62 ± 12.99	115.24 ± 13.51	0.366
Diastolic Blood Pressure (mmHg)					
Stage 1	Group A	Supplement	75.78 ± 10.23	75.61 ± 7.89	0.933
	Group B	Placebo	75.86 ± 13.35	74.05 ± 11.23	0.289
Stage 2	Group A	Placebo	77.39 ± 10.39	75.28 ±7.37	0.172
	Group B	Supplement	74.90 ± 12.63	73.90 ± 10.93	0.492
Heart Rate (beats per min)					
Stage 1	Group A	Supplement	76.39 ± 7.37	75.06 ± 7.63	0.451
	Group B	Placebo	78.52 ± 9.60	81.00 ± 8.68	0.210
Stage 2	Group A	Placebo	76.72 ± 9.14	75.33 ± 7.84	0.503
	Group B	Supplement	78.90 ± 11.56	78.86 ± 11.93	0.985
Cortisol Concentration (nmol/L)					
Stage 1	Group A	Supplement	33.85 ± 14.42	32.85 ± 18.35	0.871
	Group B	Placebo	27.70 ± 10.41	27.33 ± 14.46	0.893
Stage 2	Group A	Placebo	31.25 ± 21.70	31.82 ± 21.96	0.927
	Group B	Supplement	21.57 ± 12.92	14.11 ± 9.44	0.007 **
Total Alpha Power (uV^2^/Hz)					
Stage 1	Group A	Supplement	4.83 ± 5.31	14.95 ± 21.50	0.031 *
	Group B	Placebo	7.79 ± 11.48	8.75 ± 11.86	0.831
Stage 2	Group A	Placebo	10.22 ± 14.74	9.04 ± 12.63	0.785
	Group B	Supplement	8.75 ± 8.73	11.86 ± 12.69	0.337
Frontal Alpha Power (uV^2^/Hz)					
Stage 1	Group A	Supplement	3.53 ± 4.63	20.05 ± 34.28	0.037 *
	Group B	Placebo	6.02 ± 6.33	7.82 ± 9.73	0.381
Stage 2	Group A	Placebo	11.40 ± 15.06	10.67 ± 15.77	0.866
	Group B	Supplement	8.53 ± 9.87	14.43 ± 15.96	0.088

Significance at *p* value * indicates <0.05, ** indicates <0.01, *** indicates <0.001 (*p* > 0.05).

**Table 3 nutrients-14-00652-t003:** Crossover summed up variables after the supplement RLX2™ or placebo administration.

Variables	Pre Mean ± SD	Post Mean ± SD	Paired *t*-Test *p*-Value	Pre Mean ± SD	Post Mean ± SD	Paired *t*-Test *p-V*alue	Value Changes ± SD		Student *t*-Test *p* Value
	Supplement			Placebo			Supplement	Placebo	
PSQI Total Score	7.79 ± 2.57	5.40 ± 2.12	<0.001 ***	6.82 ± 2.20	6.13 ± 2.41	0.096	2.44 ± 1.94	0.69 ± 2.54	0.001 **
Total Sleeping Hours	5.72 ± 1.25	6.47 ± 1.11	<0.001 ***	5.83 ± 0.98	6.16 ± 1.12	0.053	−0.76 ± 0.96	−0.33 ± 1.03	0.063
C1 (Subjective Sleep Quality)	1.54 ± 0.51	1.08 ± 0.66	<0.001 ***	1.38 ± 0.63	1.15 ± 0.67	0.037 *	0.46 ± 0.68	0.23 ± 0.67	0.135
C2 (Sleep Latency)	1.43 ± 0.97	0.948 ± 0.86	<0.001 ***	1.23 ± 0.90	1.03 ± 0.90	0.088	0.49 ± 0.68	0.21 ± 0.73	0.083
C3 (Sleep Duration)	1.59 ± 0.88	1.08 ± 0.66	<0.001 ***	1.41 ± 0.79	1.331 ± 0.80	0.421	0.51 ± 0.64	0.10 ± 0.79	0.014 *
C4 (Habitual Sleep Efficiency)	0.82 ± 0.94	0.26 ± 0.55	<0.001 ***	0.51 ± 0.64	0.49 ± 0.76	0.872	0.56 ± 0.88	0.03 ± 0.99	0.013 *
C5 (Sleep Disturbances)	1.31 ± 0.47	1.21 ± 0.61	0.253	1.33 ± 0.58	1.21 ± 0.52	0.230	0.10 ± 0.55	0.13 ± 0.66	0.852
C7 (Daytime Dysfunction)	1.13 ± 0.66	0.82 ± 0.56	0.001 **	0.95 ± 0.56	0.95 ± 0.76	1.000	0.31 ± 0.52	0.00 ± 0.69	0.029 *
Systolic Blood Pressure (mmHg)	117.54 ± 12.64	118.15 ± 13.17	0.701	120.00 ± 14.47	115.59 ± 16.33	0.021 *	−0.62 ± 9.94	4.41 ± 11.44	0.042 *
Diastolic Blood Pressure (mmHg)	75.31 ± 11.45	74.69 ± 9.56	0.601	76.56 ± 11.95	74.61 ± 9.54	0.088	0.62 ± 7.28	1.95 ± 6.94	0.410
Heart Rate (beats per min)	77.74 ± 9.51	77.10 ± 10.23	0.687	77.69 ± 9.31	78.38 ± 8.68	0.626	0.64 ± 9.87	−0.69 ± 8.80	0.531
Cortisol (nmol/L)	27.24 ± 14.81	22.76 ± 16.94	0.158	29.34 ± 16.46	29.40 ± 18.19	0.984	4.48 ± 19.43	−0.62 ± 19.56	0.307
Total Alpha Power (uV^2^/Hz)	6.95 ± 7.53	13.28 ± 17.11	0.022 *	8.91 ±12.95	8.68 ±11.25	0.920	−6.33 ± 16.03	0.23 ± 14.11	0.066
Frontal Alpha Power (uV^2^/Hz)	6.23 ± 8.21	17.01 ± 25.78	0.008 **	8.49 ± 11.37	9.13 ± 12.75	0.777	−10.78 ± 23.25	−0.63 ± 13.49	0.070

Significance at *p-*value * indicates <0.05, ** indicates <0.01, *** indicates <0.001.

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
