# Peer review of "Effect of Alpha-S1-Casein Tryptic Hydrolysate and L-Theanine on Poor Sleep Quality: A Double Blind, Randomized Placebo-Controlled Crossover Trial"

_nutrients, 2022, doi:10.3390/nu14030652_

Round 1
Reviewer 1 Report
This manuscript was well-prepared. The introduction supports the study design and the conclusions tell a concise story. I wish that more graphs depicting the data had been included instead of tables however.
Reviewer 2 Report
Dear Authors:
I reviewed the manuscript entitled “Effect of Alpha-S1-Casein Tryptic Hydrolysate and L-Theanine on Poor Sleep Quality: A Double Blind, Randomized Placebo-Controlled Crossover Trial”. It is an interesting manuscript with a good concept and a better approach to insomnia than using synthetic medications. However, the basic material that was derived and used (Alpha-S1-Casein Tryptic Hydrolysate) comes from a natural food, which is milk. As a Food Scientist, I know that it will not have an adverse effect on human body. The following minor corrections are recommended.
- The sentences 35-38 are not clear and may want to split it into two sentences.
- The sentence 50 “These medications are also might be inappropriate….please change it to “These medications may also be inappropriate ….”.
- In sentences 64-65, the dose used for rats was 300 mg/kg/day that translates into 21 grams for an average man, which is a large dose. It may still be OK for humans?
- The sentences 94-96, can be changed to “…at the same time other data were collected in a quiet setting”.
5. In the sentence 157, you may want to add “…and samples with very low levels of salivary cortisol, an indicator of chronic insomnia [27].
